# Impact of benzodiazepine use on the risk of occupational accidents

**François-Olivier Baudot**[1,2]*

1 ERUDITE, Université Paris-Est Créteil, Créteil, France, 2 Caisse Nationale de l'Assurance Maladie, Paris, France

* francois-olivier.baudot@u-pec.fr

## Abstract

Benzodiazepines (BZDs) are drugs commonly used for treating insomnia and anxiety. Although they are known to induce cognitive and psychomotor impairments, their effect on the risk of causing accidents at work remains understudied. The objective of this study is to estimate this risk by differentiating between the recommended use and overuse of these drugs (i.e., uninterrupted use for four months). The data come from the French National Health Data System, which provide a population composed of French people who had at least one work accident (WA) from 2017 to 2019 (approximately 2.5 million people). A linear probability model with two-way fixed effects is used to deal with time-constant heterogeneity and the time effect independent of individuals. The results show a reduction in the risk of WA after a short period of BZD use (one month) compared with no use at all, but the risk of WA increases when treatment exceeds the recommended duration. The intensity of use results in a greater risk of WAs: a 1% increase in BZD use (expressed as the amount reimbursed) leads to a 4.4% (p<0.001) increase in the monthly risk of WAs. Moreover, we see an increase in risk in the month following the treatment discontinuation (+3.6%, p<0.001), which could be due to rebounding and catch-up effects. Health professionals and BZD users should be made aware of the WA risk induced by the use of BZDs, particularly after prolonged use and after discontinuation of treatment. This study provides more evidence for the need to limit the duration of BZD treatment.

## Introduction

The prevention of work accidents (WAs) and illnesses is a public health priority and one of the key objectives of the EU Strategic framework on health and safety at work for 2021–2027 [1]. In 2016, the global burden of work-related diseases was estimated by a joint report of the World Health Organization (WHO) and the International Labor Organization (ILO) to be 1.9 million deaths (19% from WAs) and 90 million disability-adjusted life years (DALYs) (30% from WAs) [2]. The European Agency for Safety and Health at Work (EU-OSHA) estimates (using a DALY methodology) the global cost of work-related accidents and illnesses at €2,680 billion (which is 3.9% of global GDP) and the European cost at €476 billion (which is 3.3% of the European GDP) [3]. In France, more than one million WAs were identified by the national

**Data Availability Statement:** Publicly sharing SNDS data is forbidden by law according to the French national data protection agency (Commission Nationale de l'Informatique et des Libertés, CNIL). To request data access please

contact the National Health Data Hub www.health-data-hub.fr.

**Funding:** FOB's doctoral contract was subsidized by the ANRT (Association Nationale de la Recherche et de la Technologie). Agreement n° 2017/1799. The funder had no role in study design, data collection and analysis, decision to publish, or preparation of the manuscript.

**Competing interests:** The author has declared that no competing interests exist.

health insurance fund (*Caisse Nationale de l'Assurance Maladie*, CNAM) in 2018, including commuting accidents. The frequency of WAs has decreased overall since 2000, and the slight increase in the number of WAs from 2013 to 2019 came from the increase in salaried employment [4]. In France, prevention is also a priority of the fourth *Plan santé au travail* (Health at Work Plan, PST4) [5] and requires increasing our knowledge of the causes of accidents.

Improving WAs prevention requires understanding their determinants. The economic literature on work absences and the risk of accidents at work is extensive and sheds light on the roles of cost of absence, individual characteristics, and job characteristics [6]. The role of health and health care on WAs has been understudied and may appear ambiguous due to possible mixed intertemporal effects. Poor health is likely to increase the WA risk: effect per se (in case of impairment of vision for instance), and effect of medical treatment (affecting awareness, cognition, and behavior) [7]. However, there is a strong two-way causality between health and care use, and WAs. Poor health will decrease the probability of being at work [8] and thus the WA risk. Appropriate access to care requires time off work but may result in improving health and preventing future WAs.

This study is focused on drug consumption. One particular class of drugs deserves our attention: the benzodiazepines (BZDs), which are used broadly as anxiolytics and hypnotics. Short-acting BZDs are used as sleeping pills (hypnotic BZDs), while long-acting BZDs are used as anxiolytics (anxiolytic BZDs). As the risk of dependence increases with the duration of treatment, the French National Authority for Health (HAS) recommends limiting the treatment duration to four weeks for hypnotic BZDs and twelve weeks for anxiolytic BZDs [9–11]. In France, despite a slight downward trend since 2000, 13% of the population consumed these drugs at least once in 2015, and the country remains a major consumer of BZDs in Europe [9]. The effects of BZDs may lead to a higher risk of WAs, as they have sedative and cognitive effects [12]. However, other mechanisms may also be involved (such as health improvement and reduction of occupational exposure), and the overall effect remains understudied.

Symptoms of diseases may be directly related to an increased risk of WAs. For instance, insomnia is related to impaired work performance and a higher risk of WAs [13]. More widely, chronic health problems [14] and mental health problems [15] are associated with a higher risk of WAs. On the other hand, ill health may result in unemployment. Work absences will increase during the year following the occurrence of cancer, and the employability of workers suffering from cancer decreases over time [8]. Moreover, men suffering from anxiety and men and women suffering from depression are less likely to remain in their jobs [16]. Absence from work should lead to a decrease in the WA rate.

Beyond the direct effect of diseases, medical treatments may be associated with WAs. Kaestner and Grossman [17] show that drug use could increase the WA risk for men. Some psychotropic medications could also increase the WA risk [7]. Among them, BZDs are frequently used to treat insomnia and anxiety. We may expect a positive effect on the risk of WAs because of their adverse effects. They can cause cognitive impairment [12], increase the risk of falling in the elderly [18,19], and lead to behavioral disorders [20]. The risk of traffic accidents after consuming BZDs is well known [21–23]. In France, 3.4% of traffic accidents may be related to medication intake, and half of them to BZD intake [24].

In a literature review about the role of health problems and drug treatments in accidental injury at work, Palmer et al. [7] reported that most of the studies screened suffered important limitations. The two major sources of limitation were a non-independent assessment of exposures and outcomes, and reverse causation. The increase in BZD use after a WA has been shown previously [25]. Many of these studies are cross-sectional and prone to both biases: exposures and outcomes being self-reported at the same time and with a lack of clarity about whether exposure preceded injury or followed it. Similarly, Garbarino et al. [26] consider that

the risk of confounding is high in most studies because important potential confounders are not assessed, and there is a lack of rigor in collecting information on medication use.

To assess the role played by the BZDs in accidents, sometimes patients are tested (blood or urine) after an accident without any evidence of BZD use being higher than in the general population [27–29]. This approach allows declarative bias to be avoided, which is common in cases of psychotropic drug use (including BZDs) [30,31].

Some studies found no effect of BZD intake on WA risk, such as the study by Montastruc et al. [32], who hypothesizes that knowledge of risk might provide an incentive to avoid BZD use. Gilmore et al. [33] do not find a significant association between hypnotics (in the 30 days before the date of injury) and injuries for the entire population of a health maintenance organization in the US. Other studies show a positive relationship between BZD use and increased WA risk. Voaklander et al. [34] report an OR = 3 of injury among farmers who took sedatives within 30 days before the date of injury.

Palmer et al. [7,15] provide a case-control study based on a large cohort in the UK to address some biases; the effect of consumption in the 12 months before the injury is significant for antidepressants (OR = 1.26, p<0.05) and hypnotics (OR = 1.72, p<0.001). Garbarino et al. [26] provide a meta-analysis, which do not suggest any increased risk of occupational injury among BZD users.

This study aims to determine the impact of past BZD use on WA risk, by distinguishing different levels of use according to official recommendations. Data come from the French National Health Data System (SDNS), and the study population is the French private-sector salaried population with at least one WA reported from 2017 to 2019.

## Material and methods

### Data

The study rely on data from the French National Health Data System (*Système national des données de santé*, SNDS). The SNDS contains individual data used for billing and reimbursement [35]. National Health Insurance is mandatory for all people living in France (French and foreigners). The database includes information related to outpatient health care consumption (such as physician consultation and drug reimbursement); hospital data (private and public); compensated days off work (due to sick leave, maternity, WA or occupational disease); and information related to long-term diseases, which open access to specific health care reimbursement (ALD: *Affections de longue durée*). More information is available for people whose WA leads to at least four days off work. These data are relative to the employee (such as the contract type), the employer, and the accident.

Continuous variables (expenses, number of consultations, and number of days) are capped at the 99[th] percentile (of monthly and non-zero values). Statistical analyses were performed using SAS 9.4 on Red Hat Enterprise Linux Server 7.4. Data extraction was carried out between November 18 and 19, 2020. The author did not have access to information that could identify individual participants during or after data collection.

### Population and scope of the study

The study population is insured by the general scheme, which covers mainly private-sector employees and their relatives, except for farmers. The relative WA data that were available pertained to almost 76% of the population living in France in 2015 [35]. The inclusion criteria are as follows: having experienced at least one WA from 2017 to 2019 and being between 16 and 79 in 2017. In this age group and during these years, all the WAs that occurred in France are included, except for farmers, civil servants, and self-employed workers.

The period of study is 36 months (i.e., from January 2017 to December 2019). The studied population stood at 2,544,237 at the start of 2017 (deceased people are excluded in the month following the death). In France, the National Health Insurance compensates for commuting accidents as regular WAs. In this study, WAs include accidents that occurred at work (workplace accidents) and accidents that occurred between residence and work or between work and catering area (commuting accidents).

## Econometric strategy

Panel data are gathered by calendar month, thus covering 36 periods. The model used for estimation is a linear probability model. The dependent variable is the monthly WA probability. The independent variables are related to healthcare utilization in the previous four months; the first four periods are therefore not used for estimation. Individual and monthly fixed effects are used in the model to control, respectively, variables linked to individuals but which do not vary over time and seasonal/temporal variables independent of individual characteristics.

The econometric model is written as follows:

$$WA_{it} = \beta_0 + \beta_1 BZD_{it} + \beta_2 ChronicCond_{it} + \beta_3 \log\left(1 + \sum_{\tau=t-4}^{t-1} DrugExp_{i\tau}\right) + \beta_4 Drug_{it}$$

$$+ \beta_5 \sum_{\tau=t-4}^{t-1} Consult_{i\tau} + \beta_6 \sum_{\tau=t-4}^{t-1} DaysAbs_{i\tau} + \alpha_i + \lambda_t + u_{it}$$

with t = 5, 6. . ., 36 (because of the four-month lag variables).

$WA_{it}$ is a dummy variable indicating whether the individual $i$ had a WA in month t. All control variables refer to the four months before t:

- $BZD_{it}$ is a qualitative variable with four mutually exclusive terms: no BZD use (no BZD reimbursed over the last four months), recent use (at least one BZD reimbursed in month t-1), past use (at least one BZD reimbursed in months t-4 to t-2 and none in t-1), overuse (at least one BZD reimbursed each month from t-4 to t-1).

- $ChronicCond_{it}$ is a vector of two dummies indicating whether the individual $i$ has been recognized as having a chronic psychiatric disease or any other kind of chronic disease (ALD) during the four preceding months.

- $DrugExp_{i\tau}$ is a vector of three variables indicating the amount reimbursed for psycholeptics (antipsychotics, anxiolytics, and hypnotics, excluding BZDs), antidepressants, and other reimbursed medicines; the amount used in the model is the sum (+1) from t-4 to t-1, and the log is used to normalize the distribution.

- Because there is a lot of 0 in drug reimbursements, 1 is added to each variable in $DrugExp_{i\tau}$ vector. To correct the bias induced, three dummies indicating whether the sum of drug expenses over four months is 0 are added and noted in the vector $Drugs_{it}$.

- $Consult_{i\tau}$ is a vector of three variables indicating the number of consultations with a GP (including home visits), a psychiatrist, or another specialist, summed over four months.

- $DaysAbs_{i\tau}$ is a vector of two variables indicating the number of days hospitalized and the number of compensated days off work (due to sickness, maternity, WA, or occupational disease), summed over four months.

$\alpha_i$ is the vector of individual fixed effects (i.e., differences between individuals stable over time), $\lambda_t$ is the vector of time fixed effects (i.e., time variations independent of individuals), and $u_{it}$ contains unobserved time-varying factors dependent on individuals.

Data are relative to reimbursed medications but do not provide information regarding actual drug intake; thus, this study equates *BZD use* to *BZD reimbursement*, i.e., dispensation in a pharmacy. *BZD overuse* corresponds to exceeding the treatment duration recommended by the HAS for anxiolytic BZDs: 12 weeks at most [10].

All variables in the model control for healthcare consumption. They are proxies for health conditions, potential adverse effects of drugs, and absences from work. Mental health-related variables (chronic psychiatric diseases, psychiatrist consultations, antidepressants, and other psycholeptic treatments) allow for controlling confounding factors associated with BZD reimbursements and identifying the proper effect of BZD use, as far as possible. In particular, antidepressants are distinguished because they are frequently co-prescribed with BZDs. Other drugs variable controls for potential adverse effects of drugs. Chronic conditions and doctor consultations are proxies of poor health. Doctor consultations may also refer to prevention advice and care. Days hospitalized and compensated are proxies of poor health, but above all are nonwork days, without exposition to WA risk.

Control variables are relative to the four months preceding month t. This choice is made for the sake of consistency with the period considered for BZD use. The maximum treatment duration recommended by health authorities for BZDs is 12 weeks, therefore, four months without treatment interruption are considered as overuse and then other care consumption are controlled during that time. It aims to identify the effect of past care consumption on current WA risk. It allows for avoiding simultaneity by using lagged variables, as the occurrence of a WA probably leads to a strong increase in care consumption.

Another estimation is made using a different variable of interest. In the main results, BZD overuse is defined regarding the HAS guidelines for anxiolytics BZDs. Although this recommendation is built on medical evidence, the threshold is debatable, and it may be interesting to look at the effect of an increase in BZD use. For this purpose, the qualitative BZD use variable is replaced with two variables: the log of the amount reimbursed for BZDs over the last four months (+1), which expresses the intensity of treatment, and a dummy variable, which equals 1 when there was a least one BZDs reimbursement in the same period. This dummy corrects the bias introduced by adding 1 to the amount reimbursed and expresses the effect of using BZDs with the lowest intensity, i.e., one single use.

The coefficient of time-constant factors cannot be estimated because of the use of fixed effects. Analyses were stratified to observe whether BZD use has heterogeneous effects in some specific subpopulations.

Because of the small share of the study population with more than one WA throughout the 3-year study period (14%), the lagged-dependent variable was not included in the model. Nevertheless, a robustness check was made by restricting modeling to the population with a single WA throughout the study period.

Two-way fixed effects are especially helpful where individual social and economic data are lacking. Individual fixed effects can control for certain unobserved variables (such as social origin). However, fixed effects cannot deal with unobserved variables that vary over time and between individuals. Employment is considered a fixed effect (because it is known only at the time of WA), but it could change in the study period. A robustness check is made by comparing two subpopulations: with a fixed-term contract (expected to change jobs more often) and with a permanent contract (expected to change jobs less often) at the time of WA.

Given the high number of fixed effects, estimation is very demanding on computing resources. To remedy this, a linear probability model (less computationally intensive) is used

**Table 1. Yearly WA statistics.**

| Year | N population (January) | Variable | N of people experiencing accident | % | % of the population with WA in the year |
|------|------------------------|----------|-----------------------------------|---|------------------------------------------|
| 2017 | 2,544,237 | WA | 944,859 | 37% | 100% |
| | | Workplace accident | 823,433 | 32% | 87% |
| | | Commuting accident | 127,989 | 5% | 14% |
| 2018 | 2,543,018 | WA | 953,075 | 37% | 100% |
| | | Workplace accident | 826,997 | 33% | 87% |
| | | Commuting accident | 132,700 | 5% | 14% |
| 2019 | 2,540,810 | WA | 944,342 | 37% | 100% |
| | | Workplace accident | 817,108 | 32% | 87% |
| | | Commuting accident | 133,898 | 5% | 14% |

Note: In 2017, 944,859 people experienced at least one WA (37% of the population). Eighty-seven percent of them experienced a WA in the workplace, and 14% experienced a commuting accident.

Field: study population alive in January from 2017 to 2019 (deceased people are excluded from the study population the month following the death).

instead of a non-linear estimate. A robustness check is carried out using logit estimation on a sample of one tenth of the population.

## Results

### Work accidents and healthcare use in the population

The sample comprises 2,544,237 people in January 2017. The mean age was 37 at this date (SD 12.4) and 42% were women. This is younger and more masculine than the French population in 2017 when the average age was 41 and 52% were women [36,37]; and refers to a double selection effect: people with a WA are employed and more likely to work in hazardous jobs. The age distribution is available in S1 Fig. Most of the study population is between 20 and 60 years old (92%), and the frequency of age decreases slightly from 20 to 60 years old, with a drop between 36 and 44 years old, which is in line with the literature about the age of WA [38,39].

Statistics regarding the WAs in the study population are presented for each year in Table 1. Thirty-seven percent of the population experienced at least one WA every year from 2017 to 2019. Accidents at the workplace constitute the majority of WAs (gathering workplace and commuting accidents) and affected 87% of the population experiencing a WA every year, compared to commuting accidents affecting 14% of the same population every year. Throughout the 3-year study period, 85.6% of the population experienced one single WA, 11.8% experienced two WAs, and only 2% experienced three or more WAs.

Statistics of care use related to variables used in the regressions are presented in Table 2 (dummy variables) and Table 3 (continuous variables). Statistics are given monthly, i.e., for the study population (N = 2,544,237) and at 36 months (minus the deceased people after their death) and over the 3-year study period, on average for the whole study population. Every month 3.3% had a WA, 3.4% used BZDs, and 0.9% overused them (i.e., used them for at least four months), on average. Throughout the 3-year study period, 100% had a WA (this is the selection criterion), 27% used BZDs at least once, and 3.7% overused them at least once. Other statistics refer to control variables.

Table 3 shows statistics of care use for continuous variables. The dispersion is important, especially for amounts reimbursed. We see that there are many zeros, especially in monthly values. Psycholeptics and antidepressants are little used but they are included in the model

**Table 2. Percentage of work accidents and benzodiazepine use.**

| Variables | Monthly | | 3-year | |
|---|---|---|---|---|
| | Frequency | % | Frequency | % |
| WA | 83,498 | 3.28% | 2,544,237 | 100.00% |
| BZD use | 86,824 | 3.41% | 687,544 | 27.02% |
| BZD overuse | 22,227 | 0.87% | 93,571 | 3.68% |
| Chronic psychiatric disease | 68,101 | 2.68% | 78,390 | 3.08% |
| Other chronic condition | 218,948 | 8.61% | 256,832 | 10.09% |

Note: In average, 83,498 people experienced a WA each month in the study population (i.e. 3.28%). One hundred percent of the population experienced a WA throughout the study period (3 years).

Field: Population having had at least one WA from 2017 to 2019 (N = 2,544,237).

because they are frequently used along BZDs and may be confounding factors. The high number of compensated days comes partly from illness absences due to work accidents.

## Estimations

Table 4 shows the results of WA risk estimation. Variables are added progressively to the model. In the first two specifications (without control variables, without and with fixed effects), all modalities of BZD use are associated with a decrease in WA risk (reference: no use). As expected, the effect size decreases when adding fixed effects and control variables. In the comprehensive model (specification 5), the effect of recent use remains negative (-0.18 pp (percentage point), p<0.001), and the effects of past use (+0.12 pp, p<0.001) and overuse (+0.07 pp, p<0.05) become positive on the WA risk. Relative to the probability of WA every month in the study population (3.28%), that means a decrease of 5.4% in the risk for recent BZD use, and increases of 3.6% and 2.3% for past use and overuse, respectively.

Chronic diseases (ALDs) are the variables with the largest effect on WA probability: -2.2 pp (-67.0%, p<0.001) and -1.1 pp (-33.3%, p<0.001) for chronic psychiatric diseases and other chronic diseases, respectively. Regarding drug uses (except BZDs), a 1% increase in other psychotropics use decreases the probability of WA by 0.09 pp (2.7%, p<0.001), and a 1% increase in antidepressant and other drug uses increases the risk of WAs by 0.13 pp (4.1%, p<0.001) and 0.06 pp (1.7%, p<0.001), respectively. Other drugs use also has a positive effect on the risk

**Table 3. Statistics of care use, monthly and over the 3 years.**

| Variables | Monthly | | | | 3-year | | | |
|---|---|---|---|---|---|---|---|---|
| | Mean | Q3 | P90 | Std Dev | Mean | Q3 | P90 | Std Dev |
| Mean amount reimbursed for psycholeptics (except BZDs) (€) | 0.29 | 0 | 0 | 6.43 | 10.54 | 0 | 1.84 | 186.94 |
| Mean amount reimbursed for antidepressants (€) | 0.28 | 0 | 0 | 1.98 | 10.20 | 0 | 8.03 | 50.02 |
| Mean amount reimbursed for other drugs (€) | 10.25 | 6.84 | 29.09 | 28.30 | 368.60 | 383.38 | 870.74 | 666.74 |
| Number of GP consultations | 0.34 | 0 | 1 | 0.77 | 12.36 | 17 | 27 | 11.70 |
| Number of psychiatrist consultations | 0.01 | 0 | 0 | 0.22 | 0.54 | 0 | 0 | 4.68 |
| Number of other consultations | 0.08 | 0 | 0 | 0.31 | 2.75 | 4 | 7 | 3.89 |
| Number of compensated days off work | 2.49 | 0 | 5 | 8.64 | 89.49 | 99 | 257 | 152.97 |
| Number of hospitalized days | 0.08 | 0 | 0 | 1.51 | 3.00 | 1 | 5 | 14.04 |

Note: The average amount reimbursed for psycholeptics is €0.29 a month.

*Q3 is the third quartile of the distribution and P90 is the 90th percentile.*

Field: Population having had at least one WA from 2017 to 2019 (N = 2,544,237).

**Table 4. Regressions of WA probability, variables added progressively.**

| Variables | Specification (1) | Specification (2) | Specification (3) | Specification (4) | Specification (5) |
|---|---|---|---|---|---|
| *BZDs (ref. no use)* | | | | | |
| Overuse | -0.00277*** (0.0002) | -0.01241*** (0.00031) | -0.00181*** (0.00032) | 0.00294*** (0.00022) | 0.00075* (0.00032) |
| Recent use | -0.00542*** (0.00013) | -0.01087*** (0.00015) | -0.00326*** (0.00016) | -0.00004 (0.00013) | -0.00178*** (0.00016) |
| Past use | -0.00322*** (0.0001) | -0.00731*** (0.00012) | -0.00051*** (0.00012) | 0.0021*** (0.00011) | 0.00118*** (0.00012) |
| *Chronic conditions* | | | | | |
| Psychiatric | - | - | -0.02748*** (0.00058) | 0.00146*** (0.00013) | -0.02198*** (0.00058) |
| Other diseases | - | - | -0.01526*** (0.00031) | 0.00005 (0.00007) | -0.01091*** (0.00031) |
| *Drugs reimbursed* | | | | | |
| No other psycholeptics | - | - | -0.00284*** (0.00034) | -0.00207*** (0.00025) | -0.00258*** (0.00034) |
| Other psycholeptics (log(€)) | - | - | -0.00141*** (0.00016) | -0.00053*** (0.00009) | -0.00089*** (0.00016) |
| No antidepressants | - | - | 0.00155*** (0.00043) | 0.00034 (0.00034) | 0.00228*** (0.00043) |
| Antidepressants (log(€)) | - | - | -0.00062*** (0.00016) | 0.00069*** (0.00012) | 0.00133*** (0.00016) |
| No other drugs | - | - | 0.00176*** (0.0001) | 0.00197*** (0.00008) | 0.00326*** (0.0001) |
| Other drugs (log(€)) | - | - | -0.00009** (0.00003) | 0.00084*** (0.00002) | 0.00055*** (0.00003) |
| *Doctor consultations* | | | | | |
| GP | - | - | -0.00434*** (0.00001) | -0.00115*** (0.00001) | -0.00272*** (0.00001) |
| Psychiatrist | - | - | -0.00109*** (0.00004) | -0.0001*** (0.00003) | -0.00004 (0.00004) |
| Other specialists | - | - | -0.00532*** (0.00003) | -0.00151*** (0.00003) | -0.00191*** (0.00003) |
| *Absence from work* | | | | | |
| Compensated days off work | - | - | - | -0.00024*** (0.00000) | -0.00037*** (0.00000) |
| Hospitalization days | - | - | - | -0.00019*** (0.00001) | -0.0002*** (0.00001) |
| *Fixed effects* | | | | | |
| Individual | No | Yes | Yes | No | Yes |
| Time | No | Yes | Yes | No | Yes |
| $R^2$ | 0.000037 | 0.010497 | 0.012731 | 0.002212 | 0.014474 |

Field: Population having had at least one WA from 2017 to 2019 (N = 2,544,237).

Note: * $p < 0.05$, ** $p < 0.01$, *** $p < 0.001$. Standard errors in parentheses.

-: The variable is not used in this specification.

Interpretation: In specification 1, BZD overuse (compared to no BZD use, calculated for months t-4 to t-1) is associated with a decrease of 0.277 pp of WA probability in month t.

of WAs. Medical consultations are negatively associated with the WA risk (except for psychiatrists, whose effect is not significant): an additional GP consultation in the last four months reduces the probability of WA by 0.27 pp (8.3%, p<0.001) (by 0.19 pp, 5.8%, p<0.001 for consultations with other specialists). Last, one additional day of absence from work within the

previous four months leads to a decrease of 0.04 pp (1.1%, p<0.001) and 0.02 pp (0.6%, p<0.001) in the probability of WA for compensated days and hospitalization days, respectively.

In the following estimation, instead of using a BZD use variable referring to health authorities' guidelines, a variable indicating the intensity of BZD use and a dummy variable of use are introduced in the model. Table 5 points out findings using these new BZD variables. We see

**Table 5. Regression of WA risk on the intensity of BZD use.**

| Variables | Specification (6) |
|---|:---:|
| *BZDs* | |
| BZD use (log(€)) | 0.00144***<br>(0.00015) |
| BZD use (ref. no use) | -0.0015***<br>(0.00021) |
| *Chronic conditions* | |
| Psychiatric | -0.02225***<br>(0.00058) |
| Other diseases | -0.01094***<br>(0.00031) |
| *Drugs reimbursed* | |
| No other psycholeptics | -0.00271***<br>(0.00034) |
| Other psycholeptics (log(€)) | -0.001***<br>(0.00016) |
| No antidepressants | 0.00189***<br>(0.00043) |
| Antidepressants (log(€)) | 0.00104***<br>(0.00017) |
| No other drugs | 0.00323***<br>(0.0001) |
| Other drugs (log(€)) | 0.00055***<br>(0.00003) |
| *Doctor consultations* | |
| GP | -0.00272***<br>(0.00001) |
| Psychiatrist | -0.00007<br>(0.00004) |
| Other specialists | -0.00191***<br>(0.00003) |
| *Absence from work* | |
| Compensated days off work | -0.00037***<br>(0) |
| Hospitalization days | -0.0002***<br>(0.00001) |
| *Fixed effects* | |
| Individual | Yes |
| Time | Yes |
| $R^2$ | 0.014471 |

Field: Population having had at least one WA from 2017 to 2019 (N = 2,544,237).

Note: * $p < 0.05$, ** $p < 0.01$, *** $p < 0.001$. Standard errors in parentheses.

Interpretation: An increase of 1% in BZD use in the four previous months is associated with an increase of 0.144 pp of WA probability in month t.

that the dummy's effect is negative: -0.15 pp (-4.6%, p<0.001) in the WA probability; but the effect of the intensity variable is positive: a 1% increase in BZD use leads to a 0.14 pp (+4.4%, p<0.001) increase in the risk of WAs. The results are unchanged for other variables. Among all drugs used, the effect of an additional 1% of BZD use is the largest, compared to antidepressants +0.1 pp (+3.2%, p<0.001) and other drugs +0.05 pp (+1.7%, p<0.001).

## Heterogeneous effects

To test the heterogeneity, analyses are stratified by gender, age, and duration of work interruption following the WA. Both sex and age are related to WA risk and BZD use [4,9]. The statistics show that the study population is younger and more masculine than the general French population. BZD consumption in the study population is related to sex and age. Over the 3 years, 21% of men consumed at least one BZD vs. 35% of women; 2.8% and 4.9% overused it, respectively. The frequencies of use (overuse) over the same period for people under 30, between 30 and 44, between 45 and 59, and over 60 are 18% (1.1%), 29% (3.7%), 34% (6.3%), and 33% (7.1%), respectively.

Regressions by sex show no difference compared to regressions for the whole population (see S1 Table), except for BZD overuse among women, which is no longer significant. Regarding regressions by age, the effect of overuse is positive on the risk of WAs among people under 60 years of age, but significant only between 30 and 44 (see S2 Table). For people aged 45 to 59 years, the effect size of overuse is smaller than that of past use. For people aged 60 and over, all measures of use are associated with a lower WA risk.

The length of time off work after the WA is an indicator of the accident's severity. When stratifying by the work interruption duration following the WA, the population with a single WA within the three years of the study is used (2,169,528 people) (see S3 Table). BZD overuse is positively and significantly associated with the risk of WAs leading from 8 days (median) to 26 days (3$^{rd}$ quartile) of work interruption. Below and above, the effect of overuse is not significant. We see that for the most serious accidents, i.e., leading to more than 26 days of work interruption, the effect size of past use is greater than that of recent use: +0.2 pp (p<0.001) vs. -0.13 pp (p<0.001), respectively.

## Robustness checks

The model used is not autoregressive, i.e., the lagged-dependent variable is not included in the model. The hypothesis made is that of no effect of past WA on the current risk of WAs (except via days of absence that are controlled). To relax this hypothesis, a new analysis was conducted on the subpopulation with a unique WA throughout the three-year study period. The results are shown in S4 Table and are similar to those for the full population, except for BZD overuse whose effect is no longer significant.

In this study, employment is assimilated into a fixed effect. This hypothesis can be debated because people may change jobs during the study period, and this information is not present in the data. To try to estimate whether it could be a source of bias, estimations were repeated for the populations with fixed-term and permanent contracts. People with permanent contracts are presumed to change jobs less frequently than people with fixed-term contracts. If changing jobs is a source of bias, estimates may differ between both populations. Information about the type of employment contract is available only at the time of the accident and for a few people (when the WA results in at least four days off work, which is known within two days following the accident, and the variable still includes approximately 40% missing values). The results are similar to those for the whole study population (see S5 Table), except for the effect of recent use, which becomes insignificant for fixed-term contracts (and significant at a

5% threshold for permanent contracts), and the effect of the number of consultations with a psychiatrist, which becomes positive and significant. Most importantly, the results are identical between both contract types, except for the effects of a new psychiatric illness, which is not significant for the population with a fixed-term contract. For both populations, the effect of overuse becomes larger than that of past use: 0.3 pp (p<0.001) vs. 0.15 pp (p<0.001) and 0.53 pp (p<0.001) vs. 0.21 pp (p<0.001) for permanent and fixed-term contracts, respectively.

Due to the dichotomous nature of the explained variable, the natural choice for estimates should have been a logit estimation. Because of the large dataset and individual fixed effects, the logit is too computationally intensive. Instead, a linear probability model is used. To test the influence of this estimation choice, another estimate was made for a subpopulation (randomly restricted to one-tenth of the total population) using a logit instead of a linear model. The results are presented in S6 Table. These results are not directly comparable with the main results because odds ratios are provided (vs. marginal effects), but we can see that signs and significance are coherent with the main estimation, even though the effect of two variables becomes insignificant at the 5% threshold (BZD overuse and other psycholeptics use).

## Discussion

### Discussion of results

Compared to other studies on the impact of psychotropic drug use on the risk of WAs [7,17], this study is the first, to the best of the author's knowledge, to specifically examine the influence of the duration and treatment status (active or not) of a particular medication and therefore to allow distinguishing the effects between recommended use and overuse. Moreover, the study relies on an administrative database, which avoids declarative bias that is known to be high in the case of psychotropic drug use (and even more for overuse) and misclassification of drugs [31,40,41].

To explain the negative association between BZD use and WA risk when there is no control variable (in the stepwise approach), we can assume a lower probability of working for people with worse health (captured by BZD use), which seems to dominate the potential adverse effects of BZDs. The addition of fixed effects and control variables leads to a reduction in effect size, and we see a changeover when the number of compensated days off work during the four previous months is added: effects of BZD overuse and past use become positive. It strengthens the hypothesis of a decrease in WA exposure among people treated with BZDs, captured by the working time-related variables.

In the comprehensive model, the large effect of chronic diseases reflects the health deterioration, which probably leads to a decrease in the probability of being exposed (loss of job, decrease in working time, and decrease in exposure during work). The use of psychotropic drugs and antidepressants has opposite effects (negative and positive, respectively) on the WA risk. Whether psychotropic drugs may be beneficial in reducing the risk of WAs, antidepressants have a detrimental effect, which confirms other studies [15,42]. The positive effect of the use of other drugs could reflect adverse effects of treatments, such as effects on attention, vigilance, and motor coordination. The negative effect of doctor's consultations could come from a protective effect through prevention and recommendations on the proper use of medication (including BZDs), but also capture health deterioration. Days of absence from work (hospitalization and compensated days) are negatively related to the WA risk because they reflect a non-exposition to occupational risk.

We observe a decrease in WA risk when people use BZDs in the preceding month. Regarding the adverse effects of BZDs (such as psychomotor impairments that lead to motor vehicle accidents, falls, and fractures [18]), the expected consequence of BZD use was instead an

increase in WAs the month following use. Some hypotheses can be put forward to explain this result. First, improved health may minimize accident risk. Because BZDs are recommended for treating numerous diseases and when used under medical guidelines, their use is expected to improve patient health, they could thus decrease WA risk [43]. Second, behavioral changes: workers can anticipate the risk of accidents and, if possible, try to minimize their exposure. The prescribing doctor may remind and warn patients about the risks potentially induced by BZDs, especially those whose professions make them highly exposed. While doctors seem to have good knowledge of the adverse effects of BZDs [44], the dissemination of information could be flawed [45] and result in poor knowledge of BZDs risks among patients [46]. However, information transmission could be better for highly exposed employees. Furthermore, the government and employers conduct prevention campaigns against the use of psychoactive substances in the workplace [47], and a pictogram strongly advising against driving is present on all BZDs packaging. Another hypothesis is put forward by Montastruc et al. [32], who state that the knowledge of risk may lead to avoiding BZD consumption. In France, doctors are financially incentivized to reduce their BZD prescriptions [48], so they could choose to avoid BZD prescriptions for patients who are more exposed to the WA risk. In the same vein, Garbarino et al. [26] hypothesized that the lower BZD use in some professions (such as commercial drivers) may result in a paradoxical effect: whether these professions suffer from work-related stress and poor sleep quality, they could benefit from appropriate use of BZDs.

The WA risk increases with past BZD use, i.e., in the $4^{th}$ to $2^{nd}$ preceding months but not in the last. After an initial period of decrease in exposure when the treatment started, an employee may try to catch up on delayed work. A medical explanation is also possible, as treatment discontinuation is known to lead to a rebound effect [49], i.e., anxiety or insomnia may become worse than before treatment. After only a few weeks of treatment, withdrawal symptoms may even occur in the form of irritability, increased stress, anxiety, panic attacks, difficulty in concentration, muscular pain, and stiffness, among others [50].

The effect of BZD overuse is also positively associated with the risk of WAs. The beneficial effect of BZD use quickly declines after two weeks of treatment for hypnotic BZDs and after four weeks for anxiolytics [51]. If their effectiveness is proven for short-term treatment, long-term use is controversial (adverse effects could progressively overlap with the therapeutic effect) [52]. This hypothesis is strengthened by the estimates that include the intensity of BZD use: the use is associated with a decrease in WA risk, while this risk is positively related to the intensity of BZD use. These results stress the importance of distinguishing the single use and the intensity of use. When stratifying the population, the effect of BZD overuse remains significant and positive for people between 30 and 44 and time off work following a WA between 8 and 26 days.

These different tests and stratifications show that the effect of BZD overuse is not stable. It goes from +0.53 pp (p<0.001) among people with fixed-term contracts, who are a highly selected sample from the population, to -0.3 pp (p<0.05) among people over 60. Mostly, the effect of overuse is insignificant, especially in the logit estimation, and it should be taken with caution. Exceeding the recommendations could reveal specificities of subpopulations. After 60, the overuse of BZDs may reflect a diminution of the probability of working or returning to work after health deterioration. Conversely, among people between 30 and 44, the positive effect of overuse may show both a higher risk of WAs in this age group and a higher probability of staying at work when health deteriorates. While presenteeism (i.e., going to work when sick) is more often reported among women, it is also associated with high job demand [53]. If more risky jobs (where men are overrepresented) were associated with presenteeism that may explain why men overusing BZDs remain more often than women in exposed employment.

This study aims to fill gaps in the literature by examining the effect of BZD use and compensatory mechanisms (such as more exposed workers avoiding treatment and treated patients

reducing their occupational exposure) on the WA risk. It tries to address biases often encountered in the existing literature: the declarative bias by using an administrative database, omitted variables by using fixed effects to deal with time-constant heterogeneity, and the reverse causality by accounting for past BZD use.

## Limitations and perspectives

The database only provides information on reimbursed care, and one reimbursement is supposed to correspond to one use. The risk of error exists only for medication reimbursement, if a drug is bought and not used, and mainly for a single box drug delivery. The post-treatment period could also be affected by intermittent use, particularly in the case of sales in large packages.

Measuring WAs is not an obvious task. WAs are significantly underreported in the EU [54] and the USA [55]. Underreporting in France has been estimated at approximately 20% by the French survey *Working Conditions 1998* [56]. Thus, because of underreporting, working with administrative databases entails incorporating reporting determinants into the WA determinants. One of the major causes of underreporting is job insecurity [57]. The reporting increases for serious accidents in large companies and qualified people [56].

This study was conducted on the population affected by a WA between 2017 and 2019. This population is particular because WAs are not homogeneously distributed in the population (more men, younger, in better health, employed, and other factors). Consequently, the findings of this study cannot be applied to the whole population. Calculation of the increase of WA likelihood was done from the WA probability each month of the study population, which is higher than in the general population. Therefore, the effect size could be underestimated compared to the entire population. Although non-directly comparable (different temporality and different measurement of BZD use), the effect sizes reported by Palmer et al. [7,15] in a case-control study are higher than in this study.

Some variables highlighted in the literature review are absent from the control variables (such as socioeconomic and professional characteristics) due to data limitations. The two-way fixed effect model addresses time-constant individual heterogeneity and time variations. Most of the variables involved in the risk of WAs (and not used in the model) are assumed to be fixed throughout the study period, such as social origin, education, blue/white-collar workers, business line, and urban/rural residence. Nevertheless, time-varying variables acting on both BZD use and WA risk can be sources of bias. In particular, a change in employment during the study period could lead to a change in exposition, enterprise size, work environment, and satisfaction at work. Although not controllable with the study data, the risk of bias is estimated by comparing estimates of people with fixed-term and permanent contracts at the time of the accident, which led to very slight differences. These results can increase our confidence that contract type and job change are not a source of bias.

Similarly, occupational exposure is partially controlled by the variables (particularly compensated days off work and hospitalization days), but the data do not contain information about leave, changes in work rhythms or tasks. Occupational exposure may be affected by the use of BZDs: patients may choose to reduce their activity (which reduces the risk of WAs) and increase their work intensity again when treatment is stopped (which increases the risk of WAs). This is a direct effect of BZDs and forms part of the total effect (alongside medical consequences, such as therapeutic and adverse effects). However, BZD use and WA risk may be jointly influenced by an omitted variable (which does not involve healthcare consumption), such as a difficult personal event that leads to increased BZD consumption and decreased occupational exposure. In this case, the results would be biased downwards.

One of the main limitation of the study is the lack of knowledge about jobs characteristics. The use of another database could enable better control of the professional determinants of WAs. In particular, the SNDS could be matched to other administrative databases (such as those of the national pension insurance scheme) to improve knowledge of career paths. The monitoring of BZD consumption and its consequences must continue in a context of increased consumption following the COVID pandemic.

## Conclusion

BZDs are broadly used to treat various diseases, mainly anxiety and insomnia. They have adverse effects (psychomotor and cognitive) that one may reasonably expect to increase the risk of WAs. This study shows that the resulting effect is not trivial and varies according to the duration of treatment.

The study design does not allow for distinguishing between the direct effect of BZDs and changes in occupational risk exposure. Short-term treatment with BZDs (1 month) is associated with a decreased risk of WAs in the following month. This effect may come from a healing effect or compensatory mechanisms linked to knowledge of risks (avoidance of treatment or taking extra precautions). Treatment discontinuation is associated with an increased risk of WAs that may arise from rebound effects, withdrawal symptoms, and catch-up effects. When the treatment is prolonged over the recommendation, it becomes associated with an increase in the risk of WAs, which may come from adverse effects overriding the beneficial effects of treatment. Moreover, the risk of WAs is directly related to the intensity of treatment.

These results should help to improve medical guidelines and constitute useful information about the therapeutic benefits and adverse effects of BZDs. In particular, prescribers and BZD users should be aware of the increased risks of WA after BZD use, not only at treatment initiation but also after months of use and after treatment stops. This study provides more evidence for the need to limit the duration and intensity of BZD treatments. Prevention related to psychoactive substance use in companies should take better account of the posttreatment period.

## Supporting information

**S1 Fig. Frequency of people by age in 2017.** Field: Population having had at least one WA from 2017 to 2019 (N = 2,544,237).
(PDF)

**S1 Table. Regressions of WA risk by sex.** Field: Population having had at least one WA from 2017 to 2019 (N = 2,544,237). Note: $^*$ $p < 0.05$, $^{**}$ $p < 0.01$, $^{***}$ $p < 0.001$. Standard errors in parentheses. Interpretation: For men, BZD overuse (compared to no BZD use, calculated for months t-4 to t-1) is associated with a 0.097 pp increase in WA probability at month t.
(PDF)

**S2 Table. Regressions of WA risk by age.** Field: Population having had at least one WA from 2017 to 2019 (N = 2,544,237). Note: $^*$ $p < 0.05$, $^{**}$ $p < 0.01$, $^{***}$ $p < 0.001$. Standard errors in parentheses. Interpretation: For people under 30 years old, BZD overuse (compared to no BZD use, calculated for months t-4 to t-1) is not significantly (at a 5% threshold) associated with WA probability at month t.
(PDF)

**S3 Table. Regressions of WA risk by duration of work stoppage following the WA.** Field: Population having had a single WA from 2017 to 2019 (N = 2,170,144). Note: $^*$ $p < 0.05$, $^{**}$ $p < 0.01$, $^{***}$ $p < 0.001$. Standard errors in parentheses. Interpretation: For people whose WA did not lead to a work stoppage, BZD overuse (compared to no BZD use, calculated for months

t-4 to t-1) is not significantly (at a 5% threshold) associated with WA probability at month t.
(PDF)

**S4 Table. Regression of WA risk for the population with one single WA throughout the study period.** Field: Population having had one single WA from 2017 to 2019 (N = 2,170,144). Note: $^*$ $p < 0.05$, $^{**}$ $p < 0.01$, $^{***}$ $p < 0.001$. Standard errors in parentheses. Interpretation: BZD overuse (compared to no BZD use, calculated for months t-4 to t-1) is not significantly (at a 5% threshold) associated with WA probability at month t.
(PDF)

**S5 Table. Regressions of WA risk by type of employment contract.** Field: Population having had a single WA from 2017 to 2019, resulting in at least a 4-day work stoppage, and for whom the information is available (N = 411,519). Note: $^*$ $p < 0.05$, $^{**}$ $p < 0.01$, $^{***}$ $p < 0.001$. Standard errors in parentheses. Interpretation: For people with a permanent contract at the time of WA, BZD overuse (compared to no BZD use, calculated for months t-4 to t-1) is associated with a 0.302 pp increase in WA probability at month t.
(PDF)

**S6 Table. Logit regression of WA risk in a population sample.** Field: 10% random sample (N = 254,310). Note: $^*$ $p < 0.05$, $^{**}$ $p < 0.01$, $^{***}$ $p < 0.001$. Standard errors in parentheses. Interpretation: BZD overuse (compared to no BZD use, calculated for months t-4 to t-1) is not significantly (at a 5% threshold) associated with WA probability at month t.
(PDF)

## Acknowledgments

The author would like to thank Thomas Barnay (ERUDITE, Paris-Est Créteil University) for his helpful proofreading and advice; Pascal Jacquetin (CNAM) and the team of CNAM's statistic department for their support and advice on data use; people who discussed or had comments on the paper during public presentations: Yann Videau, Sandrine Juin, Anne-Marie Konopka and Florent Sari (ERUDITE, Paris-Est Créteil University); Philippe Petit and Tiphaine Canarelli (CNAM); Eric Defebvre (CES, Paris 1 Panthéon-Sorbonne University); Carine Franc (Inserm), Alain Paraponaris (AMSE, Aix-Marseille University); and Nicolas Debarsy (CNRS); and Sandy Tubeuf (Université Catholique de Louvain), Christine Le Clainche (LEM, Lille University), and Igor Bagayev (ERUDITE, Paris-Est Créteil University) for their valuable advice.

## Author Contributions

**Conceptualization:** François-Olivier Baudot.

**Data curation:** François-Olivier Baudot.

**Formal analysis:** François-Olivier Baudot.

**Funding acquisition:** François-Olivier Baudot.

**Investigation:** François-Olivier Baudot.

**Methodology:** François-Olivier Baudot.

**Project administration:** François-Olivier Baudot.

**Resources:** François-Olivier Baudot.

**Software:** François-Olivier Baudot.

**Supervision:** François-Olivier Baudot.

**Validation:** François-Olivier Baudot.

**Visualization:** François-Olivier Baudot.

**Writing – original draft:** François-Olivier Baudot.

**Writing – review & editing:** François-Olivier Baudot.

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
