## [Decision Letter · Decision Letter 0]

9 Nov 2023

PONE-D-23-29149Impact of benzodiazepine use on the risk of occupational accidentsPLOS ONE

Dear Dr. Baudot,

Thank you for submitting your manuscript to PLOS ONE. After careful consideration, we feel that it has merit but does not fully meet PLOS ONE’s publication criteria as it currently stands. Therefore, we invite you to submit a revised version of the manuscript that addresses the points raised during the review process.

We look forward to receiving your revised manuscript.

Kind regards,

Mabel Aoun, MD, MPH

Academic Editor

PLOS ONE

Journal Requirements:

**Additional Editor Comments:**

The research question addressed by the author is very relevant. The methodology is sound but the author is required to make several amendments especially to the structure. Please do not use « I » but rather replace this pronoun by « the author ». Please move anything that includes results to the results’ section and anything that includes interpretation of results to the discussion section.

please provide as well a point-by-point response to both reviewers.

Reviewers' comments:

Reviewer's Responses to Questions

**Comments to the Author**

1. Is the manuscript technically sound, and do the data support the conclusions?

Reviewer #1: Yes

Reviewer #2: Partly

2. Has the statistical analysis been performed appropriately and rigorously? 

Reviewer #1: Yes

Reviewer #2: I Don't Know

3. Have the authors made all data underlying the findings in their manuscript fully available?

Reviewer #1: Yes

Reviewer #2: No

4. Is the manuscript presented in an intelligible fashion and written in standard English?

Reviewer #1: Yes

Reviewer #2: No

5. Review Comments to the Author

Reviewer #1: This study adds to the knowledge base by demonstrating the increased risks of work accidents associated with the intensity of benzodiazepine use.

Abstract

“we see an increase in risk in the month following the treatment discontinuation, which could be due to rebounding and catch-up effects.” Can the author specify how much the risk increased?

The author mixed the pronouns “I” and “we” throughout the manuscript.

Introduction

(lines 109-117) The review of previous studies seems lengthy. I suggest that these paragraphs be shortened.

(lines 118-125) The description of methodology and results should be removed or move to appropriate sections.

(lines 127-129) This sentence seems unnecessary

Materials and methods

This section is written in sufficient detail.

(Line 244) cite the reference (source: Institute of Statistics, INSEE) properly.

(line 244-245) “..refers to the double selection of people who are employed and have had an accident at work.” This is a confusing statement. Please rephrase it.

(lines 249-250, line 255) Please present the results for the percentage of people who experienced WA in Table 1 in terms of % as you did in the text (39%, 40%, 39%).

(lines 252-254) “Accidents at the workplace constitute the majority of WAs (gathering workplace and commuting accidents) and affected 87% of the population experiencing a WA in 2017, compared to commuting accidents affecting 14% of the same population. Please present the results in a Table as well as in the text.

The subsection, Statistics, should fall under the results section.

Results

(line 291) explain the acronym (like pp) at its first appearance.

(lines 298-300) This reflects the health deterioration, which probably leads to a decrease in the probability of being exposed (loss of job, decrease in working time, and decrease in exposure. This is an interpretation of the results, which should better be placed in the discussion section.

There are also some other statements in the results section that should instead belong in the discussion.

Discussion

Please add suggestions for future studies.

Reviewer #2: L21: avoid using first person throughout the manuscript i.e., "me" and "I". Avoid changing from past to present tense throughout the manuscript

L23: shows a decreased risk of WA for short-term BZD use compared to what? Please specify this in the abstract.

L25: Please explain in the abstract what a "percent increase" of BZD use means. Is it a percent increase in the dose (e.g., mgs)? This should be clear.

L28-30: you state here that there is a WA risk induced by BZDs at the beginning of treatment; however, you state in L25: that there is a decreased risk up to one month. Please clarify which is correct.

L59: BZDs are also used in seizure prophylaxis and acute management. Sometimes long-term use of BZDs is required (e.g., clonazepam) and this should be addressed.

L62: You should state the effects of BZD use e.g., sedation etc.

L437: You need to define what you mean by hypnotic BZDs and anxiolytic.

L441: I am not sure intensity of use is the correct term. Is this referring to dose or duration?

L503: You should include BZD withdrawal here.

L511: This recommendation should only apply to specific indications for BZDs. Long-term use can be appropriate.

Additional Comments:

1) You should be careful using the term "overused" when referring to BZD use as there are indications where this is necessary. It is, however, longer that the recommended use for sleep or mild-moderate anxiety. The introduction should outline the guidelines for BZDs.

2) The post-treatment period could be significantly affected by intermittent BZD use. People can typically hold onto, for example, 50 diazepam, for a long period of time and still be using it.

3) There seems to be key statistical values missing in the results text (e.g., confidence intervals). This needs to be included for all results; results should not just be contained in tables.

6. PLOS authors have the option to publish the peer review history of their article (what does this mean?). If published, this will include your full peer review and any attached files.

Reviewer #1: No

Reviewer #2: No

---

## [Author Response · Author response to Decision Letter 0]

8 Dec 2023

I thank the reviewers for their analysis of the paper and very helpful comments. I describe below how I revised the manuscript in order to take systematically into account each of the reviewer’s comments. 

Additional Editor Comments:

The research question addressed by the author is very relevant. The methodology is sound but the author is required to make several amendments especially to the structure. Please do not use « I » but rather replace this pronoun by « the author ». Please move anything that includes results to the results’ section and anything that includes interpretation of results to the discussion section.

please provide as well a point-by-point response to both reviewers.

Thank you for your comment. I have modified the manuscript to never use the first person (I use passive voice or “the author” instead). I have reorganized the results and discussion sections to meet your request.

Below is the point-by-point response to the reviewers. 

Reviewers' comments:

Reviewer #1: This study adds to the knowledge base by demonstrating the increased risks of work accidents associated with the intensity of benzodiazepine use.

Abstract

“we see an increase in risk in the month following the treatment discontinuation, which could be due to rebounding and catch-up effects.” Can the author specify how much the risk increased?

I have added the value of the increase (+3.6%) in the abstract.

The author mixed the pronouns “I” and “we” throughout the manuscript.

I have replaced the first person with the passive voice throughout the manuscript. I use “we” only to comment on observations that include the reader.

Introduction

(lines 109-117) The review of previous studies seems lengthy. I suggest that these paragraphs be shortened.

I have reassembled the section on the limitations of existing studies, shortened the section on the presentation of results and merged the two paragraphs.

(lines 118-125) The description of methodology and results should be removed or move to appropriate sections.

(lines 127-129) This sentence seems unnecessary

I have deleted the corresponding sentences.

Materials and methods

This section is written in sufficient detail.

(Line 244) cite the reference (source: Institute of Statistics, INSEE) properly.

I have added the references to the bibliography.

(line 244-245) “..refers to the double selection of people who are employed and have had an accident at work.” This is a confusing statement. Please rephrase it.

I have reworded the sentence: “…and refers to a double selection effect: people with a WA are employed and more likely to work in hazardous jobs”.

(lines 249-250, line 255) Please present the results for the percentage of people who experienced WA in Table 1 in terms of % as you did in the text (39%, 40%, 39%).

(lines 252-254) “Accidents at the workplace constitute the majority of WAs (gathering workplace and commuting accidents) and affected 87% of the population experiencing a WA in 2017, compared to commuting accidents affecting 14% of the same population. Please present the results in a Table as well as in the text.

I have slightly modified the first part of the statistics section:

- I have added age distribution figure as supplementary material.

- I have modified Table 1 to focus on accident statistics. It contains the number and proportion of people experiencing the different type of accidents, as well as in the text.

The subsection, Statistics, should fall under the results section.

Modification made.

Results

(line 291) explain the acronym (like pp) at its first appearance.

Modification made.

(lines 298-300) This reflects the health deterioration, which probably leads to a decrease in the probability of being exposed (loss of job, decrease in working time, and decrease in exposure. This is an interpretation of the results, which should better be placed in the discussion section.

There are also some other statements in the results section that should instead belong in the discussion.

I have moved all the sentences interpreting the results to the discussion section.

Discussion

Please add suggestions for future studies.

I have added a paragraph at the end of the discussion.

Reviewer #2: L21: avoid using first person throughout the manuscript i.e., "me" and "I". Avoid changing from past to present tense throughout the manuscript

I have changed the use of first person to passive voice throughout the manuscript. I have also modified manuscript to consistently use the present tense, except when I am commenting on something in the past (for example, “The average age of the study population was 37 in January 2017”).

L23: shows a decreased risk of WA for short-term BZD use compared to what? Please specify this in the abstract.

I have changed the sentence: “The results show a reduction in the risk of WA after a short period of BZD use (one month) compared with no use at all”.

L25: Please explain in the abstract what a "percent increase" of BZD use means. Is it a percent increase in the dose (e.g., mgs)? This should be clear.

I have changed the sentence: “The intensity of use results in a greater risk of WAs: a 1% increase in BZD use (expressed as the amount reimbursed)”.

L28-30: you state here that there is a WA risk induced by BZDs at the beginning of treatment; however, you state in L25: that there is a decreased risk up to one month. Please clarify which is correct.

There is no contradiction between the two sentences, because one refers to the direct effect of BZDs and the other to what was observed in the study, which may also be the consequence of adaptation and risk reduction measures. The adverse effects of BZDs are likely to increase the risk of WAs, and the doctor's preventive action may be concentrated at the start of treatment and become less pronounced over time. This study shows a reduction of the risk during the month after starting to use BZDs, not necessarily from a direct effect of the treatment but rather from prevention and risk reduction. The message I would like to convey with this sentence is that preventive action against WAs is required during all the BZD treatment and even after the treatment has been discontinued, and not that doctors should stop to prevent risk when treatment starts.

I have changed the sentence to make it clearer: “Health professionals and BZD users should be made aware of the WA risk induced by the use of BZDs, particularly after prolonged use and after discontinuation of treatment.”

L59: BZDs are also used in seizure prophylaxis and acute management. Sometimes long-term use of BZDs is required (e.g., clonazepam) and this should be addressed.

In France, BZDs are only indicated for short-term use (see French National Authority for Health guidelines cited in the article). The only indication for clonazepam is the treatment of epileptic seizure.

Reference for clonazepam indication: https://www.vidal.fr/medicaments/substances/clonazepam-4157.html

L62: You should state the effects of BZD use e.g., sedation etc.

I have changed the sentence: “The effects of BZDs may lead to a higher risk of WAs, as they have sedative and cognitive effects”. I do not go into details about these adverse effects, as I am more precise two paragraphs later.

L437: You need to define what you mean by hypnotic BZDs and anxiolytic.

I have added a definition of the terms hypnotic and anxiolytic BZDs in the third paragraph of the introduction, along with recommended treatment durations:

“One particular class of drugs deserves our attention: the benzodiazepines (BZDs), which are used broadly as anxiolytics and hypnotics. Short-acting BZDs are used as sleeping pills (hypnotic BZDs), while long-acting BZDs are used as anxiolytics (anxiolytic BZDs). As the risk of dependence increases with the duration of treatment, the French National Authority for Health (HAS) recommends limiting the treatment duration to four weeks for hypnotic BZDs and twelve weeks for anxiolytic BZDs (9,16,17).”

L441: I am not sure intensity of use is the correct term. Is this referring to dose or duration?

I use the log of amount reimbursed as a proxy for the dose. This method is described in the “Econometric strategy” section: the log of the amount reimbursed for BZDs over the last four months (+1), which expresses the intensity of treatment”.

L503: You should include BZD withdrawal here.

I have added it in the conclusion: “Treatment discontinuation is associated with an increased risk of WAs that may arise from rebound effects, withdrawal symptoms, and catch-up effects.”

L511: This recommendation should only apply to specific indications for BZDs. Long-term use can be appropriate.

In France, there is no recommendation for the use of BZDs after twelve weeks. Long-term prescribing only exists outside the scope of marketing authorization. The literature cited highlights the risk of adverse effects when treatment is prolonged. See the recommendations here (in French, with abstract in English): Revet A, Yrondi A, Montastruc F. Règles de bon usage des benzodiazépines. Presse Médicale. 2018 Oct 1;47(10):872–7.

Additional Comments:

1) You should be careful using the term "overused" when referring to BZD use as there are indications where this is necessary. It is, however, longer that the recommended use for sleep or mild-moderate anxiety. The introduction should outline the guidelines for BZDs.

I have added guidelines for BZDs in the introduction. As there is no indication in France for use beyond twelve weeks, I consider longer use to be overuse, and I indicate it in the “Econometric strategy” section: “BZD overuse corresponds to exceeding the treatment duration recommended by the HAS for anxiolytic BZDs: 12 weeks at most”.

2) The post-treatment period could be significantly affected by intermittent BZD use. People can typically hold onto, for example, 50 diazepam, for a long period of time and still be using it.

I have added this in the limitation section: “The post-treatment period could also be affected by intermittent use, particularly in the case sales in large packages.”

3) There seems to be key statistical values missing in the results text (e.g., confidence intervals). This needs to be included for all results; results should not just be contained in tables.

As I provide the dispersion measures in the form of standard errors, I have added significance through the p-value in the text, for all results. For the sake of readability, I have indicated standard errors only in the tables.

---

## [Decision Letter · Decision Letter 1]

15 Feb 2024

PONE-D-23-29149R1Impact of benzodiazepine use on the risk of occupational accidentsPLOS ONE

Dear Dr. Baudot,

Thank you for submitting your manuscript to PLOS ONE. After careful consideration, we feel that it has merit but does not fully meet PLOS ONE’s publication criteria as it currently stands. Therefore, we invite you to submit a revised version of the manuscript that addresses the points raised during the review process.

As a final step before acceptance, we would like you to address the comments of the statistical reviewer.

We look forward to receiving your revised manuscript.

Kind regards,

Mabel Aoun, MD, MPH

Academic Editor

PLOS ONE

Journal Requirements:

Reviewers' comments:

Reviewer's Responses to Questions

**Comments to the Author**

1. If the authors have adequately addressed your comments raised in a previous round of review and you feel that this manuscript is now acceptable for publication, you may indicate that here to bypass the “Comments to the Author” section, enter your conflict of interest statement in the “Confidential to Editor” section, and submit your "Accept" recommendation.

Reviewer #3: (No Response)

2. Is the manuscript technically sound, and do the data support the conclusions?

Reviewer #3: Yes

3. Has the statistical analysis been performed appropriately and rigorously? 

Reviewer #3: No

4. Have the authors made all data underlying the findings in their manuscript fully available?

Reviewer #3: Yes

5. Is the manuscript presented in an intelligible fashion and written in standard English?

Reviewer #3: Yes

6. Review Comments to the Author

Reviewer #3: The manuscript could be further improved.

Line 21, the complete name of the two-way fixed effects test is to be provided.

Line 116-125, whether the database is an open-source type and information on the type of data/file format that could be extracted and accessed is to be mentioned. Source link to be provided (if any).

Line 131 -143 is to be placed before section (Line 115).

Line 146-147, the statement is unclear and requires revision.

Line 218, while fixed effects can help control for time-constant heterogeneity, they are unable to fully address variables that are completely missing from the dataset.

Line 220-222, although this approach can offer valuable insights and help address the missing data, it may not fully address potential sources of bias or confounding related to missing variables. What about the social origin and professional exposure variables (Line 217-218)? More information is to be provided/presented/discussed.

Line 225-231, more baseline characteristics are to be provided and presented in table form.

Table 2, 0 & 1 are to be denoted. Frequency is to be provided apart from %.

Table 3, Q3, and P90 are to be denoted.

Table 4, the word 'stepwise' and Line 290 'the variable is not used in this specification' is not clear. More information is to be provided in the table footnote to describe the models. Model fit indices is to be presented.

Table 5, the selection method is to be stated.

Line 282-283, the sentence requires improvement.

Decimal point for percentage figures is to be standardized.

S6 Table & S6 Table Page 31, 10th is to be denoted.

7. PLOS authors have the option to publish the peer review history of their article (what does this mean?). If published, this will include your full peer review and any attached files.

Reviewer #3: No

---

## [Author Response · Author response to Decision Letter 1]

11 Mar 2024

I thank the reviewer for his/her analysis of the paper and his/her very helpful comments. I describe below how I revised the manuscript in order to take systematically into account each of the reviewer’s comments. 

Journal Requirements:

Although I use bibliography management software, I have no information about a retracted reference in the bibliography. The only change I have made is to reference 25, because the article has been published.

Reviewer #3: The manuscript could be further improved.

Line 21, the complete name of the two-way fixed effects test is to be provided.

“Two-way FE” refers to the econometric approach used in this study rather than to a one- or two-sided test of statistical significance. I have modified the sentence to show the name of the model used: “A linear probability model with two-way fixed effects is used.”

Line 116-125, whether the database is an open-source type and information on the type of data/file format that could be extracted and accessed is to be mentioned. Source link to be provided (if any).

The database is not open-source. Publicly sharing SNDS data is prohibited by law, according to the French national data protection agency (Commission Nationale de l’Informatique et des Libertés, CNIL). Researchers can request access from the National Health Data Hub (www.health-datahub.fr). I have provided this information in the form sent to the editor (“Data availability” section).

Line 131 -143 is to be placed before section (Line 115).

It seems more logical to me to place the "population" section after the "data" section, given that the choice of population selection is based on the source of the data and that this paragraph refers to the previous one. Furthermore, looking at recent publications in PLOS ONE, it appears that the "data" section is placed before the "population" section in the articles consulted (see for example https://journals.plos.org/plosone/article?id=10.1371/journal.pone.0298068#sec002 or https://journals.plos.org/plosone/article?id=10.1371%2Fjournal.pone.0298516#sec006) 

Line 146-147, the statement is unclear and requires revision.

I have replaced the sentence with the following: “The model used for estimation is a linear probability model. The dependent variable is the monthly WA probability. The independent variables are related to healthcare utilization in the previous four months; the first four periods are therefore not used for estimation.”

Line 218, while fixed effects can help control for time-constant heterogeneity, they are unable to fully address variables that are completely missing from the dataset.

Month*year fixed effects control seasonal/temporal variables that are independent of individual characteristics. Individual fixed effects control variables that are linked to individuals but do not vary over time. These fixed effects cannot handle unobserved variables that vary over time and between individuals. If these variables are linked to both individual characteristics and the outcome, they can be a source of bias.

I have added this in the “Econometric strategy” section:

Line 148-150: “Individual and monthly fixed effects are used in the model to control, respectively, variables linked to individuals but which do not vary over time and seasonal/temporal variables independent of individual characteristics.”

Line 216-219: “Two-way fixed effects are especially helpful where individual social and economic data are lacking. Individual fixed effects can control for certain unobserved variables (such as social origin). However, fixed effects cannot deal with unobserved variables that vary over time and between individuals.”

Line 220-222, although this approach can offer valuable insights and help address the missing data, it may not fully address potential sources of bias or confounding related to missing variables. What about the social origin and professional exposure variables (Line 217-218)? More information is to be provided/presented/discussed.

I consider that social origin is fixed over time and is therefore eliminated by individual fixed effects. Job type can vary with individual characteristics and over time, and be source of bias. This is why I carry out a robustness check on two sub-populations: with and without long-term contracts (see section ‘robustness checks’). As far as occupational exposure is concerned, I consider two possibilities for the variable to affect the result:

- If there is a change in occupation exposure because of BZD use, I consider it part of the observed effect. Thus, in the ‘discussion of results’ section, I interpret the results in terms of medical effects and occupational exposure.

- If there is an omitted variable that affect both the probability of using BZDs and occupational exposure, it may bias the results.

To clarify these points, I have added a paragraph in the ‘limitations’ section.

Line 225-231, more baseline characteristics are to be provided and presented in table form.

The only demographic variables available for the entire population in the database are age and sex. The other variables relate to health care consumption. I provide descriptive statistics for discrete variables in Table 2 and for continuous variables in Table 3. Age and sex statistics are provided off-form because they are the only ones not related to care consumption. In addition, I provide the age distribution as supporting information. Unfortunately, the database does not contain socio-economics variables for the entire population.

Table 2, 0 & 1 are to be denoted. Frequency is to be provided apart from %.

I have added two frequency columns corresponding to %.

Table 3, Q3, and P90 are to be denoted.

I have added the following in the legend: “Q3 is the third quartile of the distribution and P90 is the 90th percentile.”

Table 4, the word 'stepwise' and Line 290 'the variable is not used in this specification' is not clear. More information is to be provided in the table footnote to describe the models. Model fit indices is to be presented.

I have replaced the word ‘stepwise’ with ‘variable added progressively’ in the text and in the table title. I renamed the column, so that the text, columns and footnote refer to the different specifications.

I added R2 to the bottom of the table.

Table 5, the selection method is to be stated.

In this study, the word ‘stepwise’ did not refer to a variable selection method, but to the fact that I added variables gradually. I rephrase without the word ‘stepwise’, because there is no selection in the regression presented in Table 5. 

Line 282-283, the sentence requires improvement. 

I have improved the sentence: ‘Medical consultations are negatively associated with the WA risk (except for psychiatrists, whose effect is not significant): an additional GP consultation in the last four months reduces the probability of WA by 0.27 pp (8.3%, p<0.001) (by 0.19 pp, 5.8%, p<0.001 for consultations with other specialists).’

Decimal point for percentage figures is to be standardized.

Modification made.

S6 Table & S6 Table Page 31, 10th is to be denoted.

I changed the title and footnote of the table: ‘S6 Table. Logit regression of WA risk in a population sample. Field: 10% random sample (N=254,310).’

---

## [Decision Letter · Decision Letter 2]

15 Mar 2024

PONE-D-23-29149R2Impact of benzodiazepine use on the risk of occupational accidentsPLOS ONE

Dear Dr. Baudot,

Thank you for submitting your manuscript to PLOS ONE. After careful consideration, we feel that it has merit but does not fully meet PLOS ONE’s publication criteria as it currently stands. Therefore, we invite you to submit a revised version of the manuscript that addresses the points raised during the review process.

Thank you for responding to the statistical reviewer’s comments. One final concern is the way results are reported. The author should keep in mind that this is not a thesis on statistics. We are aware that statistical analyses were performed to provide the readers with meaningful results.

1-Please avoid using the statistical technical terms in the tables. You defined your dummy variables in the methods. In the results’ sections give them a descriptive title: «Percentage of  Work accidents and benzodiazepine use? » 2-Please provide at the beginning of the results in the section that is called « statistics »once more the number of the population studied and mean and SD of age. Average should not be used in scientific reporting.The title of this section can be improved as well. We know that the author is reporting « statistics » what we would like to see is « what are the results about? 

3-Please provide R2 in Table 5 as suggested by the statistical reviewer.4-Please put descriptions instead of numbers in tables 4 and 5. The reader is not supposed to guess what is (1) (2)…et. Anything that is not clear in the tables should be defined in the footnote.5-Please remove No=1 or Yes=1 or anything similar from the columns in Tables. The author is asked to rephrase the variable in a way that reflects what he aims to. For instance: Male sex is enough to understand that it is(1). The author can put Ref. Female (to point out that the reference is 0).  

We look forward to receiving your revised manuscript.

Kind regards,

Mabel Aoun, MD, MPH

Academic Editor

PLOS ONE

Journal Requirements:

Reviewers' comments:

Reviewer's Responses to Questions

**Comments to the Author**

1. If the authors have adequately addressed your comments raised in a previous round of review and you feel that this manuscript is now acceptable for publication, you may indicate that here to bypass the “Comments to the Author” section, enter your conflict of interest statement in the “Confidential to Editor” section, and submit your "Accept" recommendation.

Reviewer #3: (No Response)

2. Is the manuscript technically sound, and do the data support the conclusions?

Reviewer #3: Yes

3. Has the statistical analysis been performed appropriately and rigorously? 

Reviewer #3: Yes

4. Have the authors made all data underlying the findings in their manuscript fully available?

Reviewer #3: Yes

5. Is the manuscript presented in an intelligible fashion and written in standard English?

Reviewer #3: Yes

6. Review Comments to the Author

Reviewer #3: The authors have put in great effort to address the comments.

Minor comment(s).

Table 5, R^2 is to be provided.

7. PLOS authors have the option to publish the peer review history of their article (what does this mean?). If published, this will include your full peer review and any attached files.

Reviewer #3: No

---

## [Author Response · Author response to Decision Letter 2]

25 Mar 2024

Thank you for responding to the statistical reviewer’s comments. One final concern is the way results are reported. The author should keep in mind that this is not a thesis on statistics. We are aware that statistical analyses were performed to provide the readers with meaningful results.

1-Please avoid using the statistical technical terms in the tables. You defined your dummy variables in the methods. In the results’ sections give them a descriptive title: «Percentage of Work accidents and benzodiazepine use? » 

I have changed the title to “Percentage of work accidents and benzodiazepine use”.

2-Please provide at the beginning of the results in the section that is called « statistics »once more the number of the population studied and mean and SD of age. Average should not be used in scientific reporting.The title of this section can be improved as well. We know that the author is reporting « statistics » what we would like to see is « what are the results about? 

I have added sample size and SD of age. I changed the section title to: “Work accidents and healthcare use in the population”.

3-Please provide R2 in Table 5 as suggested by the statistical reviewer.

I have added R2 in the table.

4-Please put descriptions instead of numbers in tables 4 and 5. The reader is not supposed to guess what is (1) (2)…et. Anything that is not clear in the tables should be defined in the footnote.

I have changed to “Specification (1)”, as in the text and in the footnote.

5-Please remove No=1 or Yes=1 or anything similar from the columns in Tables. The author is asked to rephrase the variable in a way that reflects what he aims to. For instance: Male sex is enough to understand that it is(1). The author can put Ref. Female (to point out that the reference is 0). 

I have made modifications to Tables 4 and 5.

I would like to thank reviewers and editor again for their valuable advice. I hope that the manuscript is now ready for publication in Plos One. Of course, I remain at your disposal to make any changes you may wish.

---

## [Editor Report · Decision Letter 3]

1 Apr 2024

Impact of benzodiazepine use on the risk of occupational accidents

PONE-D-23-29149R3

Dear Dr. Baudot,

We’re pleased to inform you that your manuscript has been judged scientifically suitable for publication and will be formally accepted for publication once it meets all outstanding technical requirements.

Kind regards,

Mabel Aoun, MD, MPH

Academic Editor

PLOS ONE
---

## [Editor Report · Acceptance letter]

4 Apr 2024

PONE-D-23-29149R3 

PLOS ONE

Dear Dr. Baudot, 

I'm pleased to inform you that your manuscript has been deemed suitable for publication in PLOS ONE. Congratulations! Your manuscript is now being handed over to our production team.

Kind regards, 

on behalf of

Dr. Mabel Aoun 

Academic Editor

PLOS ONE